

# Integration of exome-seq and mRNA-seq using DawnRank, identified genes involved in innate immunity as drivers of breast cancer in the Indian cohort

Snehal Nirgude[1,2], Sagar Desai[1], Vartika Khanchandani[1],
Vidhyavathy Nagarajan[1], Jayanti Thumsi[3] and Bibha Choudhary[1]

[1] Institute of Bioinformatics and Applied Biotechnology, Bengaluru, Karnataka, India
[2] Human Genetics, Children's Hospital of Philadelphia, Philadelphia, USA
[3] BGS Glenegales Hospital, Bengaluru, Karnataka, India

## ABSTRACT

Genetic heterogeneity influences the prognosis and therapy of breast cancer.
The cause of disease progression varies and can be addressed individually. To identify
the mutations and their impact on disease progression at an individual level, we
sequenced exome and transcriptome from matched normal-tumor samples.
We utilised DawnRank to prioritise driver genes and identify specific mutations in
Indian patients. Mutations in the C3 and HLA genes were identified as drivers of
disease progression, indicating the involvement of the innate immune system.
We performed immune profiling on 16 matched normal/tumor samples using
CIBERSORTx. We identified CD8+ve T cells, M2 macrophages, and neutrophils to
be enriched in luminal A and T cells CD4+ naïve, natural killer (NK) cells activated, T
follicular helper (Tfh) cells, dendritic cells activated, and neutrophils in
triple-negative breast cancer (TNBC) subtypes. Weighted gene co-expression
network analysis (WGCNA) revealed activation of T cell-mediated response in ER
positive samples and Interleukin and Interferons in ER negative samples. WGCNA
analysis also identified unique pathways for each individual, suggesting that rare
mutations/expression signatures can be used to design personalised treatment.

## INTRODUCTION

Advanced and metastatic breast cancer (BC) remain incurable. The routine treatment
regime is based on histopathology, TNM (T-tumor, N-nodes, M-Metastasis) staging, and
proliferation index according to the Ki67 and molecular subtype (*Li et al., 2020*; *Davey
et al., 2021*). Even though multi-gene expression tests such as Oncotype DX and Endo
Predict tests are implemented for (neo-)adjuvant settings of breast cancer therapy, they
are not applicable in the setting of advanced disease (*Hempel et al., 2020*). Breast cancer
has a higher complexity; hence, large-scale genomics studies are required to understand
the heterogeneity and evolution of advanced breast cancer (*Smith et al., 2019*). The
advancements in NGS technologies and the emergence of the omics field have led to the

Corresponding author
Bibha Choudhary, vibha@ibab.ac.in

**How to cite this article** Nirgude S, Desai S, Khanchandani V, Nagarajan V, Thumsi J, Choudhary B. 2023. Integration of exome-seq and
mRNA-seq using DawnRank, identified genes involved in innate immunity as drivers of breast cancer in the Indian cohort. PeerJ 11:e16033
development of various approaches to studying cancer. The common strategies to identify molecular mechanisms in cancer include scanning the genome for cancer-specific mutations and exploring the differential expression of mRNA through transcriptomics or that of protein through proteomics (*Chakraborty et al., 2018*). One of the hallmarks of cancer progression is the evasion of the immune system (*Mortezaee, 2020*; *Hanahan, 2022*; *Vinay et al., 2015*). Transcriptome analysis of the tumor tissue *vs* normal tissue using various bioinformatics analysis methods (*Zhang et al., 2020*; *Smid et al., 2016*) has led to the classification of the tumors into immunogenic and non-immunogenic cancers (*Medler et al., 2021*). The use of immune checkpoint inhibitors (ICI) has been used to treat breast cancer (*Swoboda & Nanda, 2018*; *Polk et al., 2018*), and the importance of understanding each individual immune profile would help in reducing the adverse effects of ICI treatment (*Bedognetti et al., 2016*).

Exome-seq data is used to identify variants across the exonic regions of the genome. In contrast, RNA-seq is usually used for expression profiling and to study events like splicing and RNA editing (*O'Brien et al., 2015*). An integrated analysis enables one to determine the effects of variants on gene expression, validate common mutations at the gene and transcript level or even explore variants that may no longer be observed at the transcript level due to RNA-editing. Moreover, when using only exome seq data, some variants may not be functionally relevant. Similarly, when using only RNA-seq, the underlying cause of differential expression may not be identified.

DawnRank was selected for integrating the exome and transcriptome data. DawnRank is one tool that can give an insight into driver mutations at both levels, patient and overall (*Hou & Ma, 2014*). This tool can detect driver genes from a single patient sample, working in a personalised manner. It provides a function to calculate an overall rank in the cohort. The basic concept of the ranking system in DawnRank is based on the impact of differential expression a potential driver might have on the genes connected to it downstream; that is to say, a driver can be identified by determining the effects it has on genes that are regulated by it.

Cancer incidence has been increasing in India, with more female than male cases reported in 2020 (*Mathur et al., 2020*; *Begum et al., 2021*). Among female malignancies, breast cancer is ranked first (*Toi et al., 2010*). It was also observed that the onset of breast cancer in the Indian population occurs at a much earlier age (45–50 years) than in other high-income countries (age >60 years) (*Toi et al., 2010*). A step towards understanding the molecular mechanisms governing breast cancer in Indian patients is thus required with a multi-omics approach. Also, there are no studies on the immune status of the BC subtypes in the Indian cohort.

This study is focused on understanding the molecular mechanisms that govern different breast cancer subtypes through integrating exome and transcriptome data.

## METHODS

### Study cohort and sample classification

The breast cancer patient samples used for the study were procured from BGS Global hospital, Bengaluru, Karnataka, India. The patients/participants provided their written

informed consent to participate in this study. The tumor tissue and their respective matched normal samples were collected in RNA later. Trizol was added to the samples and stored at −80 °C until further processing. Here, we analysed the exome and transcriptome data obtained from tumor samples of breast cancer patients of Indian origin to understand the underlying molecular mechanism involved. Table 1 gives the details of the patient samples used in the study. Most of the patients used in the study had Grade II tumors and infiltrating ductal carcinoma. The study was performed under ethical approval from BGS Global Hospitals and IBAB (IEC/Approval/2018-05/06/01A).

## RNA isolation and library preparation

Total RNA was extracted using the standard Trizol method from matched tumors and normal samples. RNA was quantitated using QUBIT, and quality was checked using Tapestation. mRNA libraries were prepared using Illumina TruSeq RNA Library Prep Kit v2 as described (*Nirgude, Desai & Choudhary, 2022*; *Nirgude et al., 2022*). Briefly, mRNA was isolated using oligo-dT beads and followed by fragmentation of isolated mRNA to 200–250 bp. Fragmented RNA was then converted to cDNA, followed by adaptor ligation, end repairing and PCR amplification. Size selection was performed on Adaptor ligated libraries using ampure beads. After construction of the libraries, their concentrations and insert sizes were detected using Qubit and Agilent Tapestation, respectively. High throughput sequencing was performed using Illumina HiSeq2500 to obtain 100-bp paired-end reads.

## Differential gene expression analysis

To obtain clean data, filtering was done on raw reads output from the Illumina Hiseq2500 platform. The sequencing depth for each sample was >10 million reads. As described (*Nirgude, Desai & Choudhary, 2022*; *Nirgude et al., 2022*), the quality of the data obtained was checked using the FastQC tool. The reads were aligned with Bowtie2 (*Langmead & Salzberg, 2012*) to hg38 (GRCh38) reference genome. The adapter trimming was done using trim_galore, followed by alignment with Bowtie2 (*Langmead & Salzberg, 2012*). We used Phred+33 encoding with Bowtie2 to ensure good quality of reads. The tool coverageBed from BEDTools (*Quinlan, 2014*) was used to extract the count per transcript per sample using the annotation files. This bioinformatics analysis pipeline is established as described (*Nirgude et al., 2022*). Differential expression analysis of normal and tumor patient samples was performed using the DESeq (*Anders & Huber, 2010*) R package at individual level. DESeq2 (*Love, Huber & Anders, 2014*) R package was also used to analyse differentially expressed genes in the tumor compared to normal from breast patient samples.

## Genomic DNA extraction from tumor samples

As stated before, the samples were collected in RNA Later solution. The tissue was then homogenised in Trizol reagent. After RNA extraction, DNA extraction was done using

**Table 1 Clinical and sequencing data of patients used in the study.**

| Patient ID | Sample type | T/N status | Age | ER | PR | HER | Ki67 | Grade | Type | Sample ID | mRNA-seq | | Exome-seq | |
|---|---|---|---|---|---|---|---|---|---|---|---|---|---|---|
| P1 | EPH | T | 56 | Positive | Positive | Positive | 30 | II | IDC | 1 | 20721813 | 80.73 | 64590480 | 98.03 |
| P2 | EPH | T | 35 | Positive | Positive | Positive | 50 | II | IDC | 2 | 12658236 | 90.59 | 58227244 | 92.86 |
| P3 | TNBC | T | 29 | Negative | Negative | Negative | 85 | II | IDC | 3 | 1286947 | 86.29 | 70505590 | 94.65 |
| P4 | EP | T | 41 | Positive | Positive | Negative | 20 | II | IDC | 4 | 26746372 | 83.77 | 66323748 | 96.4 |
| P5 | EPH | N | 56 | Positive | Positive | Positive | 30 | II | IDC | 1 | 56529004 | 82.58 | – | – |
| P6 | EPH | N | 35 | Positive | Positive | Positive | 50 | II | IDC | 2 | 27476976 | 60.52 | 81760000 | 80.29 |
| P7 | TNBC | N | 29 | Negative | Negative | Negative | 85 | II | IDC | 3 | 19695525 | 82.5 | – | – |
| P8 | EP | N | 41 | Positive | Positive | Negative | 20 | II | IDC | 4 | 88633702 | 65.14 | 49418862 | 83.99 |
| P9 | E | N | 62 | Positive | Negative | Negative | 40 | II | IDC | 7 | 88633702 | 80.15 | – | – |
| P10 | E | T | 62 | Positive | Negative | Negative | 40 | II | IDC | 7 | 58706131 | 87.97 | – | – |
| P11 | E | N | 41 | Positive | Negative | Negative | – | – | – | 8 | 31459697 | 78.92 | – | – |
| P12 | E | T | 41 | Positive | Negative | Negative | – | – | – | 8 | 71426900 | 89.3 | – | – |
| P14 | EH | T | 49 | Positive | Negative | Positive | 80 | II | IDC | 9 | 38179386 | 83.89 | – | – |
| P15 | EH | N | 48 | Positive | Negative | Positive | 30 | II | IDC | 10 | 34248716 | 79.99 | – | – |
| P16 | EH | T | 48 | Positive | Negative | Positive | 30 | II | IDC | 10 | 27730086 | 70.02 | – | – |
| P19 | E | N | 43 | Positive | Negative | Negative | 20 | II | IDC | 11 | 4522747 | 61.38 | – | – |
| P21 | Hmod | N | 50 | Negative | Negative | Positive | 35 | II | IDC | 12 | 34507685 | 80.28 | – | – |
| P22 | Hmod | T | 50 | Negative | Negative | Positive | 35 | II | IDC | 12 | 10037213 | 60.57 | – | – |
| P25 | Hmod | N | 60 | Negative | Negative | Positive | 50 | II | IDC | 13 | 29423645 | 50.71 | – | – |
| P26 | Hmod | T | 60 | Negative | Negative | Positive | 50 | II | IDC | 13 | 7475282 | 50.13 | – | – |
| P27 | TNBC | N | 60 | Negative | Negative | Negative | 80 | II | IDC | 14 | 23682953 | 86.26 | – | – |
| P28 | TNBC | T | 60 | Negative | Negative | Negative | 80 | II | IDC | 14 | 40932847 | 85.1 | – | – |
| P29 | E | N | 58 | Positive | Negative | Negative | 50 | – | – | 15 | 37316055 | 43.27 | – | – |
| P30 | E | T | 58 | Positive | Negative | Negative | 50 | – | – | 15 | 26337863 | 82.85 | – | – |
| 42T | EP | T | 66 | Positive | Positive | Negative | 80 | II | IDC | 16 | 39178586 | 71.17 | – | – |
| 43N | EP | N | 50 | Positive | Positive | Negative | >20 | – | – | 17 | 107610019 | 65.83 | – | – |
| 43T | EP | T | 50 | Positive | Positive | Negative | | – | – | 17 | 39004652 | 88.72 | – | – |
| 44N | EPH | N | 75 | Positive | Positive | Positive | 20 | – | – | 18 | 12516990 | 70.76 | – | – |
| 44T | EPH | T | 75 | Positive | Positive | Positive | | – | – | 18 | 27060531 | 86.13 | – | – |
| 45N | EPH | N | 56 | Positive | Positive | Positive | 30 | II | IDC | 5 | 52613399 | 58.86 | – | – |
| 45T | EPH | T | 56 | Positive | Positive | Positive | 30 | II | IDC | 5 | 13718450 | 8250.00% | – | – |
| 46N | TNBC | N | 41 | Negative | Negative | Negative | 60 | II | IDC | 6 | 49228756 | 82.88 | – | – |
| 46T | TNBC | T | 41 | Negative | Negative | Negative | 60 | II | IDC | 6 | 151518191 | 86.89 | – | – |

Note:
N, normal; T, tumor; ER, estrogen receptor; PR, progesterone receptor; HER2, human epidermal growth factor receptor2; EPH-ER, PR, HER2 positive; TNBC, triple negative breast cancer negative for ER, PR, HER2; EP-ER, PR positive; Hmod-HER2 positive; E-ER positive; IDC, infiltrating ductal carcinoma.

Back Extraction Buffer (BEB) (*Bridges Lab, 2023*). Briefly, BEB was added to trizol tubes containing only the interphase and organic (lower) phase of samples after RNA extraction. DNA was precipitated using isopropanol and 70% ethanol. DNA samples were then dissolved in the TE buffer.

## Exome-seq library preparation and sequencing

A total of ~200 ng of genomic DNA isolated from Indian breast cancer patient samples was used as input for library preparation. 260/280 ratio for each sample was calculated, and samples with a ratio of 1.8–2.0 were chosen. dsDNA fragments with 3′ or 5′ overhangs of 150–200 bp (peak size) were generated using Covaris. End repair was done using T4 DNA polymerase and T4 polynucleotide kinase enzyme to generate blunt ends. A total of 3′ ends were then adenylated to prevent them from ligating one another during the adapter ligation reaction. After adapter ligation, enrichment of the DNA library was done (*Agilent Technologies, 2021*; *Pon & Marra, 2015*). After construction of the libraries, their concentrations and insert sizes were detected using Qubit and Agilent Tapestation, respectively. High throughput sequencing was performed using Illumina HiSeq2500 to obtain 100-bp paired-end reads.

## Exome variant calling

The quality of the data obtained was checked using the FastQC tool. The reads were further processed, aligned and variants were called as detailed in the study mentioned (*Desai et al., 2021*). Briefly, the reads were aligned with Bowtie2 (*Langmead & Salzberg, 2012*) to hg38 (GRCh38) reference genome and SAMtools (*Li et al., 2009*) was used for obtaining BAM files. Picard tools (*Broad Institute, 2023b*) were used to remove PCR duplicates and the Mutect2 module of GATK (Genome Analysis Toolkit, Broad Institute, Cambridge, MA, USA) was used for variant calling (*DePristo et al., 2011*). We also performed variant calling using pileup utilities from BCFTools (*Li et al., 2009*). Variants common to both approaches were annotated using the SnpEff (*Cingolani et al., 2012b*) and SnpSift tools (*Cingolani et al., 2012a*).

## Integrated exome and RNA-seq

After analysing the application of different tools identified from the literature, DawnRank was selected for integrating the exome and transcriptome data (*Hou & Ma, 2014*). This tool can detect driver genes from a single patient sample, working in a personalized manner and it provides a function to calculate an overall rank in the cohort. The basic concept of the ranking system in DawnRank is based on the impact of differential expression a potential driver might have on the genes connected to it downstream; that is to say, a driver can be identified by determining the effects it has on genes that are regulated by it. The tool requires three inputs a gene-interaction network, somatic mutation profile, and the differential expression profile. With these three inputs, the method can rank genes by their impacts on the genes connected to it directly or indirectly by considering their expression. The gene network is viewed as a directed graph, and a random walk approach, which does so iteratively is used, like the one in PageRank. Here, a node can either walk randomly to a downstream node with a probability d (damping factor or go back to the same node with a probability) 1-d, thus symbolizing the impact on the connected genes. This approach considers the network and the differential expression of its downstream genes and ranks mutated genes with their potential to be a driver. The impact score of a gene will be higher if it's highly connected to downstream genes that are

differentially expressed. The overall rank of a driver across patients is found by aggregating the rankings in individual patients, using the Condorcet method of voting. In this method, pairwise comparisons are made between genes and the one having a higher impact score is placed at a higher rank. As DawnRank assigns scores to all genes irrespective of the presence of a mutation, the voting evaluation is done only for those pairs of genes where one of them is mutated to avoid the comparison between non mutated genes.

### Immune profiling using CIBERSORTX

To determine the average immune cell fractions in all the samples using their transcriptome profiles, paired tumor/matched normal samples were used. The immune cell fractions were subsequently compared across the samples using the online tool CIBERSORTx (*Stanford University, 2023*), which has a predetermined set of signature genes for every immune cell type and their standard expression values known as the Lm22 matrix. On uploading transcriptome data into the software, it is compared to the Lm22 matrix, and the percentage of immune cell fraction in every sample is determined.

### Identifying pathways specific to each subtype using weighted gene co-expression network analysis (WGCNA)

To correlate gene expression data of samples with their respective subtypes, we used the R package WGCNA (*Langfelder & Horvath, 2008*). It starts by constructing a gene co-expression network followed by identifying gene modules based on hierarchical clustering. It correlates these modules to the clinical data/parameters provided by the user and provides correlation values along with the significance of the correlation. The user can then select the most significantly correlated parameter-module pair and proceed with pathway analysis or network analysis of the genes in the module.

## RESULTS

### Transcriptome analysis of breast cancer subtypes

Here we analysed the data for samples which had both exome-seq and mRNA-seq data. More than 12 million reads with ~60–90% alignment were obtained for mRNA-seq, and more than 29 million reads with ~84–98% alignment were obtained for exome-seq (Table 1). Tumor and normal samples clustered separately in Principal component analysis (Fig. 1A). Interestingly, the TNBC tumor segregated separately from other tumor samples. Using DEseq (*Anders & Huber, 2010*; *DESeq, 2023*), a total of 982, 101, 102 and 843 differentially expressed (DE) genes were obtained for P1, P2, P3 and P4 samples respectively (log 2 fold change > 1.5 $p$ value < 0.05; Fig. S1). To identify common DE genes between tumor samples, the union of DE genes of all the samples was performed (Fig. 1B). The four samples when compared together did not show any common gene. However, two samples (P1, P2) that belonged to the same subtype (ER, PR and HER2 positive) shared the maximum common genes (102), indicating subtype-specific gene expression. We also performed differential gene analysis of all tumor samples together compared to normal using DESeq2 (*Love, Huber & Anders, 2014*).

Further to correlate mutations with expression using DawnRank, exome-seq data was analysed for mutations, and a mutation matrix was created taking all protein-coding

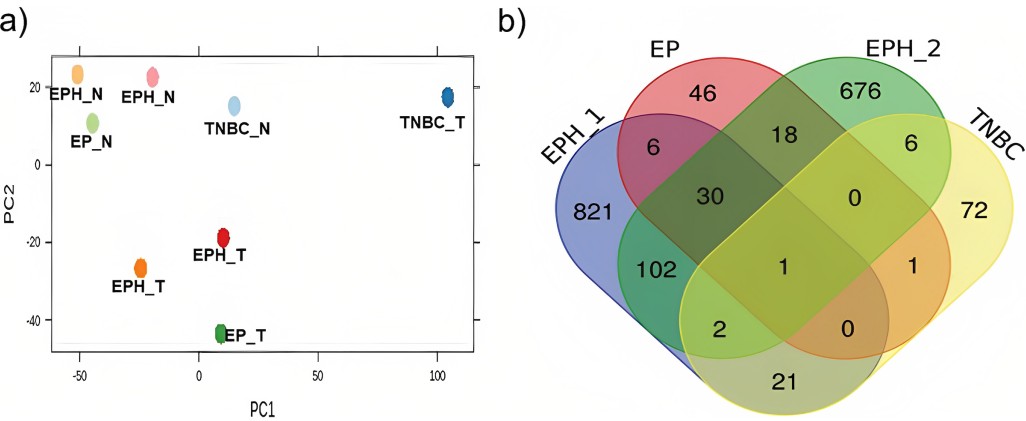

**Figure 1 Segregation of breast cancer samples based on differential gene expression.** (A) PCA plot for normal and tumor samples from Indian breast cancer patients (EPH- ER, PR, HER2 +ve; EP-ER, PR +ve; N-normal; T-tumor) (B) Venn diagram for DE genes from all four Indian breast cancer patient samples.

mutations. RNA-seq data was processed, and the counts were normalised (TPM), the network data used was the one provided by the authors of DawnRank.

## Identification of the driver genes in BC patients by DawnRank

The initial analysis was based on the top 60 genes, based on the F1 score, and the precision and recall measures for the Top N genes were reported (*Hou & Ma, 2014*). These genes were classified based on their molecular function, shown in Fig. 2A, depicted by the pie chart. Most of the gene products are involved in binding; this includes various cell surface receptors (ex. *HLA-A*), nuclear receptors (ex. *HNF1A*) and kinase activators (ex. CCDN1). They were followed by those involved in catalytic activities such as *CDKN1B*, a cyclin-dependent kinase inhibitor. The next category of genes consists of transcription regulators and transducers, including *EPAS1, POU2F1, TRRAP, HNF1A, HNF4A, PPARG* and *STAT6*. Of these top 60 genes in aggregate result, 26 genes were previously reported in the Cancer Gene Census, of which 11 were present in the gold standard list provided in the tool. These genes and their ranks are shown in Fig. 2B, where the novel drivers that might have a critical role in breast cancer are represented in blue.

Exome seq analysis gave a list of different types of mutations summarised in Fig. 3 along with expression summary for the same set of genes. Further variants obtained from the exome analysis for the 60 genes were plotted to depict types of mutations (Frameshift, missense, *etc.*) represented in Fig. 4. The changes in the individuals' expression and mutational burden were observed in genes related to the innate immune system and cell cycle.

## Most mutated genes belong to the innate immune system and cell cycle related pathways

DawnRank ranks mutated genes based on their potential to be a driver gene in cancer. Here, we analysed the top 60 genes for overlapping pathways using GSEA (*Broad Institute, 2023a*), for which the overlap matrix is shown in Fig. 5. The maximum

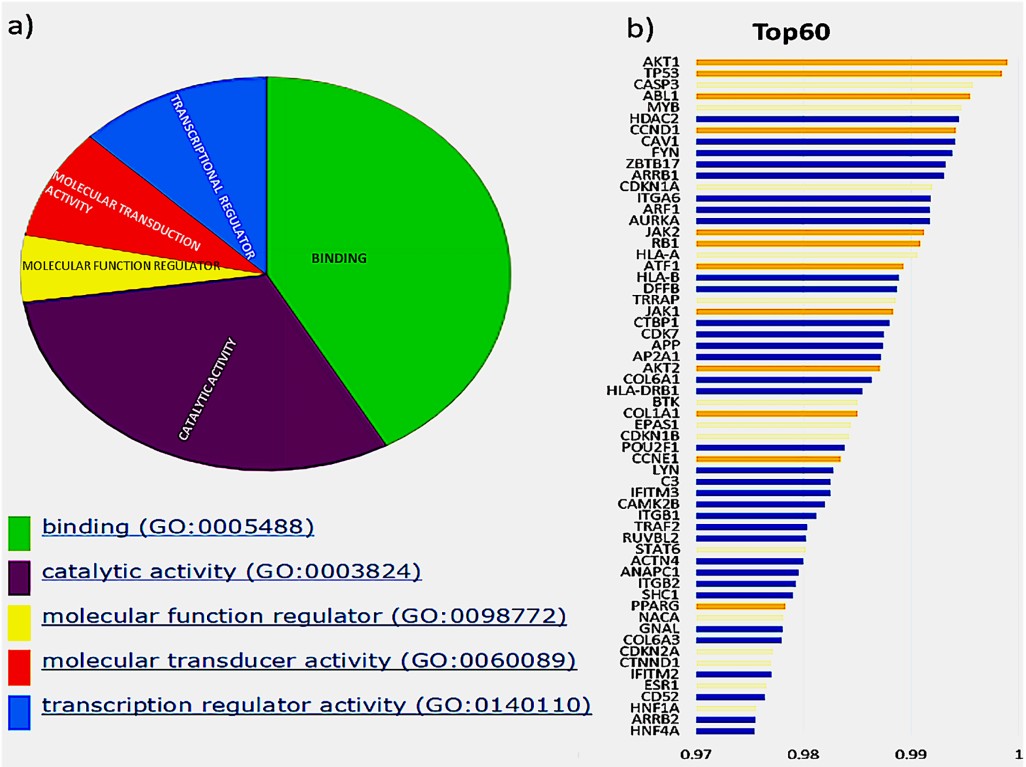

**Figure 2** **Top 60 cancer driver genes and their annotation.** (A) A pie chart representing GO annotation results from breast cancer transcriptomics for different molecular functions of the gene products. (B) Top 60 genes with their ranks (Gold, gold standard; Yellow, CGC; Blue, Novel), of the 26 CGC genes, 11 are gold standard represented in gold and CGC in yellow, the novel drivers that might have a critical role in breast cancer are represented in blue.            

overlaps were found to be cytokine signaling in the immune system reported in Reactome, followed by Pathways in Cancer reported in KEGG. It was evident that most of the genes were involved in cancer, and their function was known, showing that DawnRank could identify cancer drivers with analysed prior data. Since most of these pathways involved either the innate immune system or the cell cycle, agreeing to the hallmarks of cancer, the mutations these genes carried, and their expression were further studied.

### High frequency somatic variant in HLA-A, HLA-B, And HLA-DRB1

The human version of the major histocompatibility complex (MHC), also known as the Human Leukocyte Antigen (HLA) complex, consists of a group of related proteins that help the immune system to distinguish "self" from "non-self" peptides. The major MHC Class I genes in humans consist of *HLA-A, HLA-B* and *HLA-C*, whereas the major MHC Class II genes include *HLA-DP, HLA-DQ* and *HLA-DR*. Three of the HLA genes, *HLA-A, HLA-B* and *HLA-DRB1*, were ranked in the top 60 genes and had missense variants across all four samples, as shown in the plot (Fig. 4A). The presence of these somatic variants in all the patients suggested its strong implication in the adaptive or innate immune response to breast cancer. One of the HLA-A gene variants (rs41559716) resulted in the change of glutamine at the 78[th] position with an arginine residue in three patient samples, and its

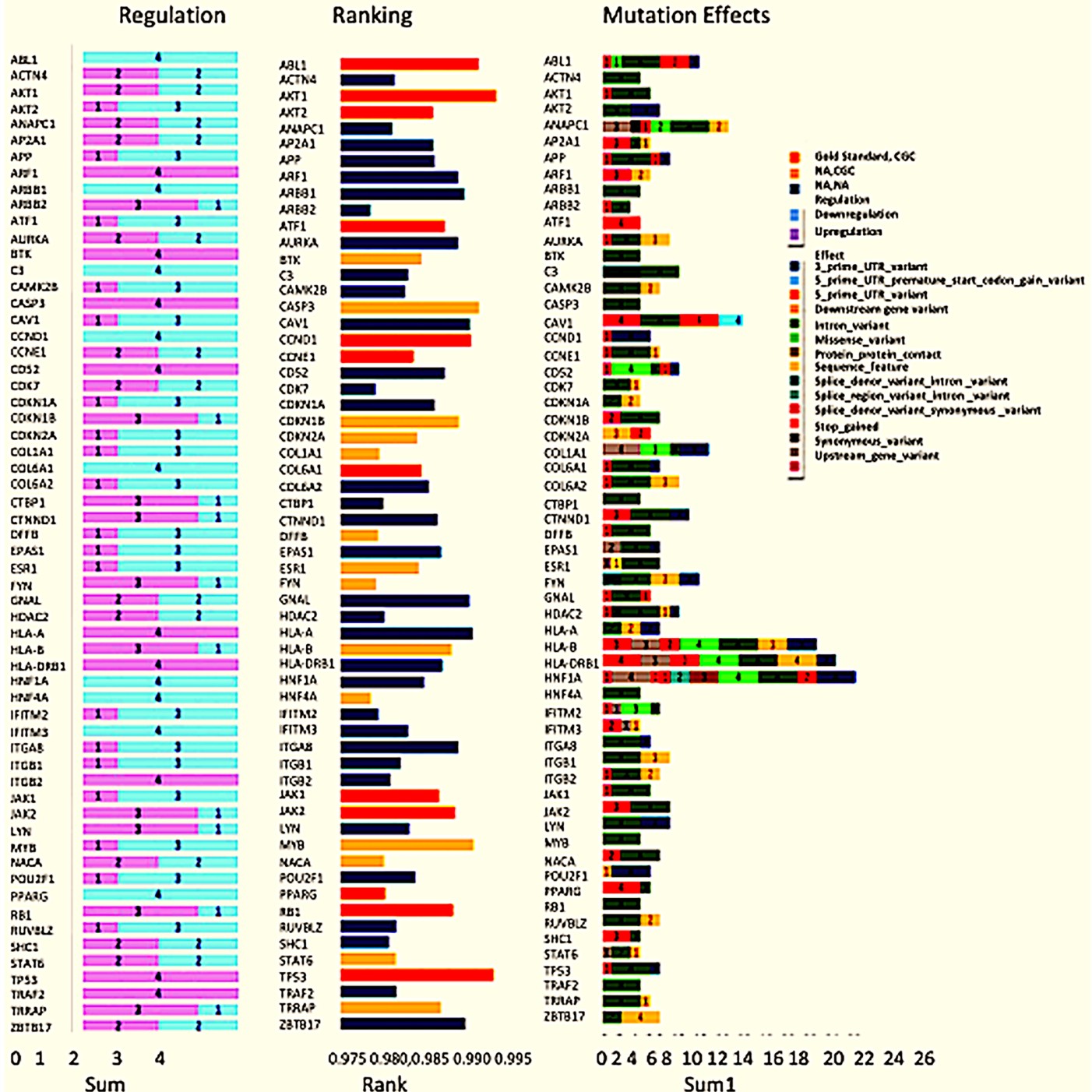

**Figure 3 Expression and mutation status of the top 60 driver genes in breast cancer.** The bar chart under the regulation column represents the number of patients where the gene is upregulated and down-regulated. Similarly, the bar chart under the Mutation effects column represents the number of patients with that type of mutation. The graph under the rankings column shows the rankings as given by DawnRank.

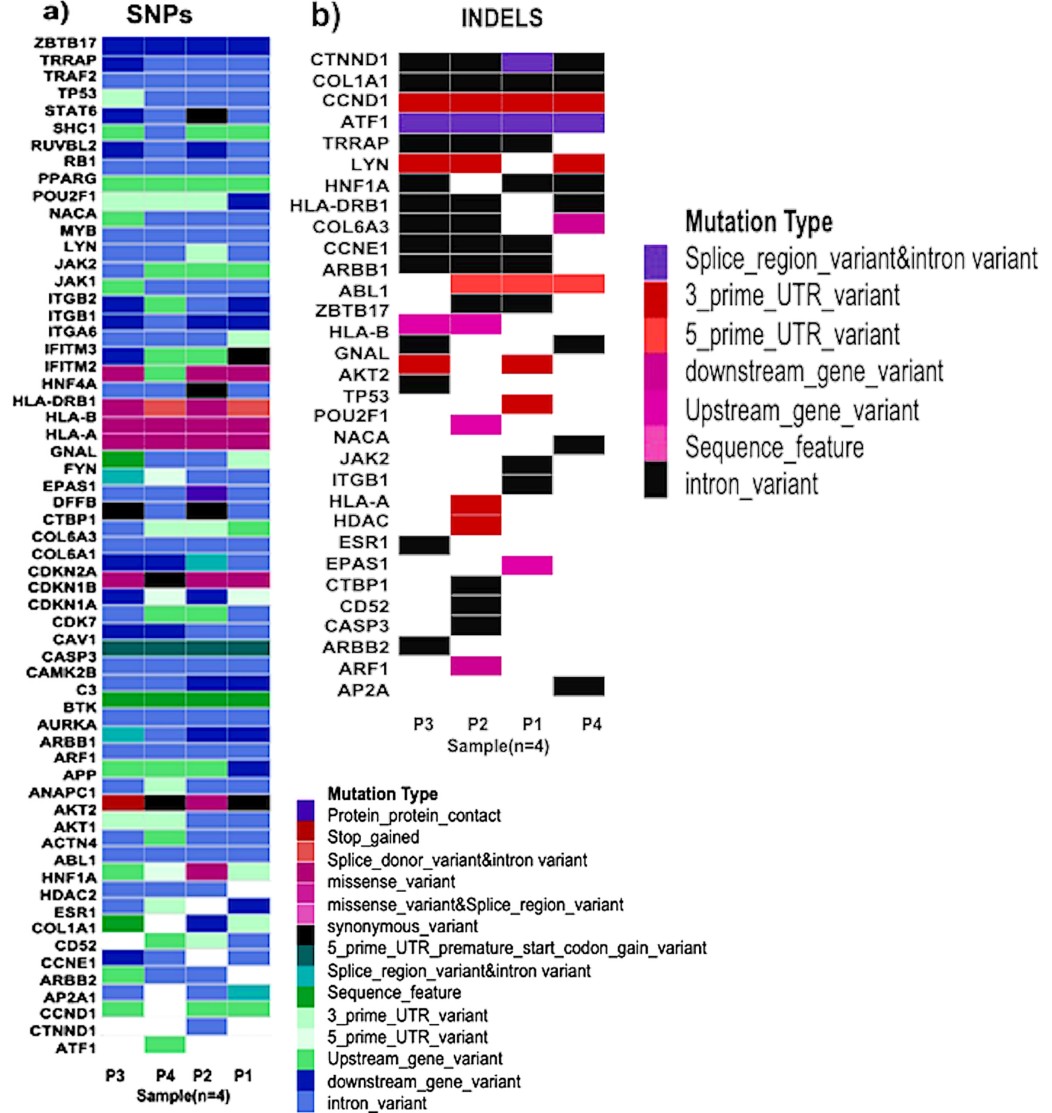

**Figure 4 Types of mutations.** (A) Patterns of SNPs in each patient sample of the cohort analysed by whole exome sequencing. Each column represents one tumor sample. Each colour represents the type of mutation. (B) Patterns of INDELs in each patient sample of the cohort analysed by whole exome sequencing. Each column represents one tumor sample. Each color represents the type of INDEL.

functional impact was assessed to be probably damaging by *PolyPhen-2 (2021)* (Fig. 6A). Three amino acid changes, R106L (rs3180379), G107R (rs3180380) and W191S (rs1050692) in HLA-B (Fig. 6B), were present in 75% of the patients of which G107R and W191S substitutions were predicted to be deleterious by Mutation Assessor with low confidence. Similarly, HLA-DRB1 (Fig. 6C), a class II MHC gene, had many variants. The most prominent mutation was SNP id rs9269942. This missense variant substituted an alanine residue at the 100[th] aa with glutamic acid/threonine and was also predicted to be deleterious.

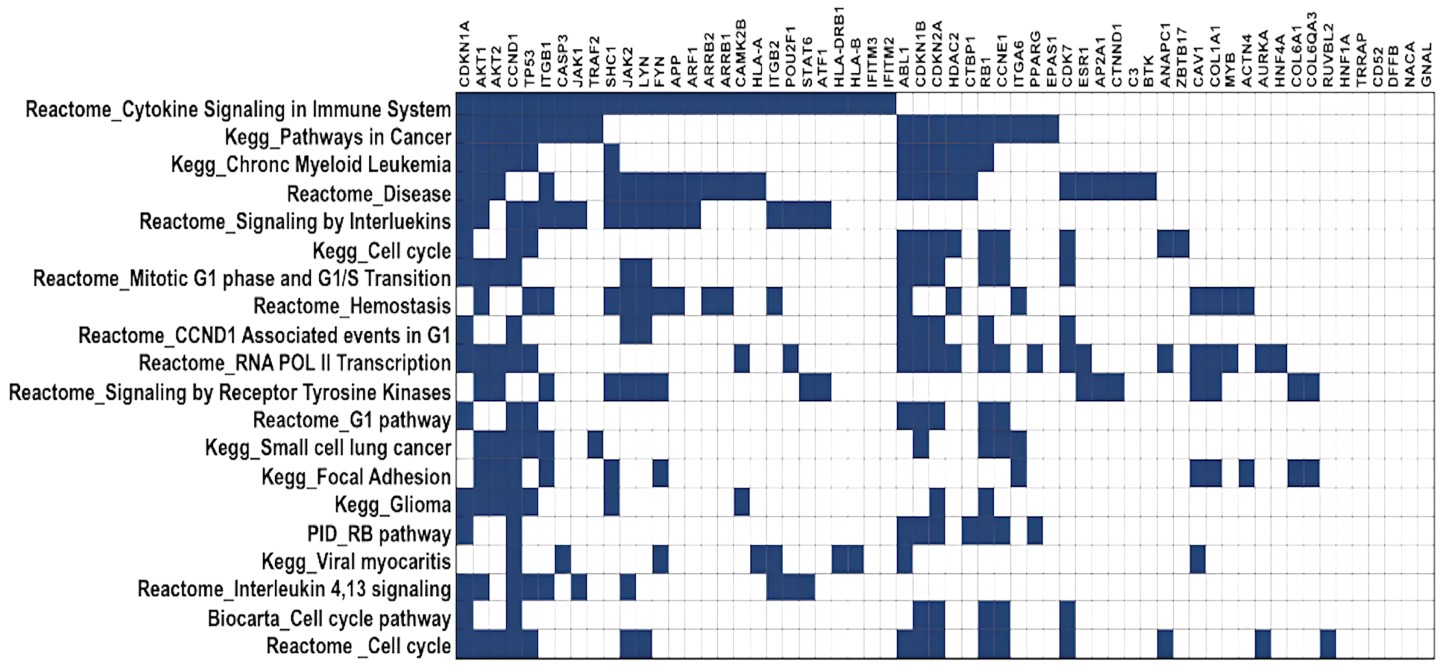

**Figure 5  Gene overlap matrix for top 20 pathways reported in KEGG, reactome, biocarta, and PID.** Cell cycle, immune regulation and cancer are the pathways enriched in mutations.

The HLA-A mutation Q78R belongs to the MHC class I alpha chain domain, as reported in *Pfam (2021)*, which could render the binding non-functional or affect the affinity. Subsequently, the mutations in HLA-B and HLA-DRBQ1 are also located in the binding domain of MHC Class I and II, respectively.

### C3 downregulated in all patients

C3 was reported to be downregulated (Fig. 3) and possessed a somatic variant in the intron region having SNP id rs366510 in all four patient samples (Fig. 3). Complement activation is a well-known immunosurveillance mechanism against cancer, the three pathways, although having a different initiating event, end up with a convertase complex that cleaves the C3 complement into C3a and C3b, further contributing to the formation of the Membrane Attack Complex (MAC). Initially thought only to kill antibody-coated tumor cells, this system has now been demonstrated to play roles in tumor progression, making its effect contradicting and increasing complexity (*Zhang et al., 2019*; *Revel et al., 2020*). It has also been suggested that the complement system's effect depends on the composition of the tumor microenvironment, the site of activation and the sensitivity of tumor cells toward the complement system (*Roumenina et al., 2019*; *Revel et al., 2020*).

### Recurrent hotspot mutation in CDKN2A

CDKN2A, a cyclin-dependent kinase inhibitor involved in the cell cycle, was reported to have a somatic variant identified by rs200429615. It results in a missense variant that is reported to be a recurrent hotspot in a population-scale cohort of tumor samples
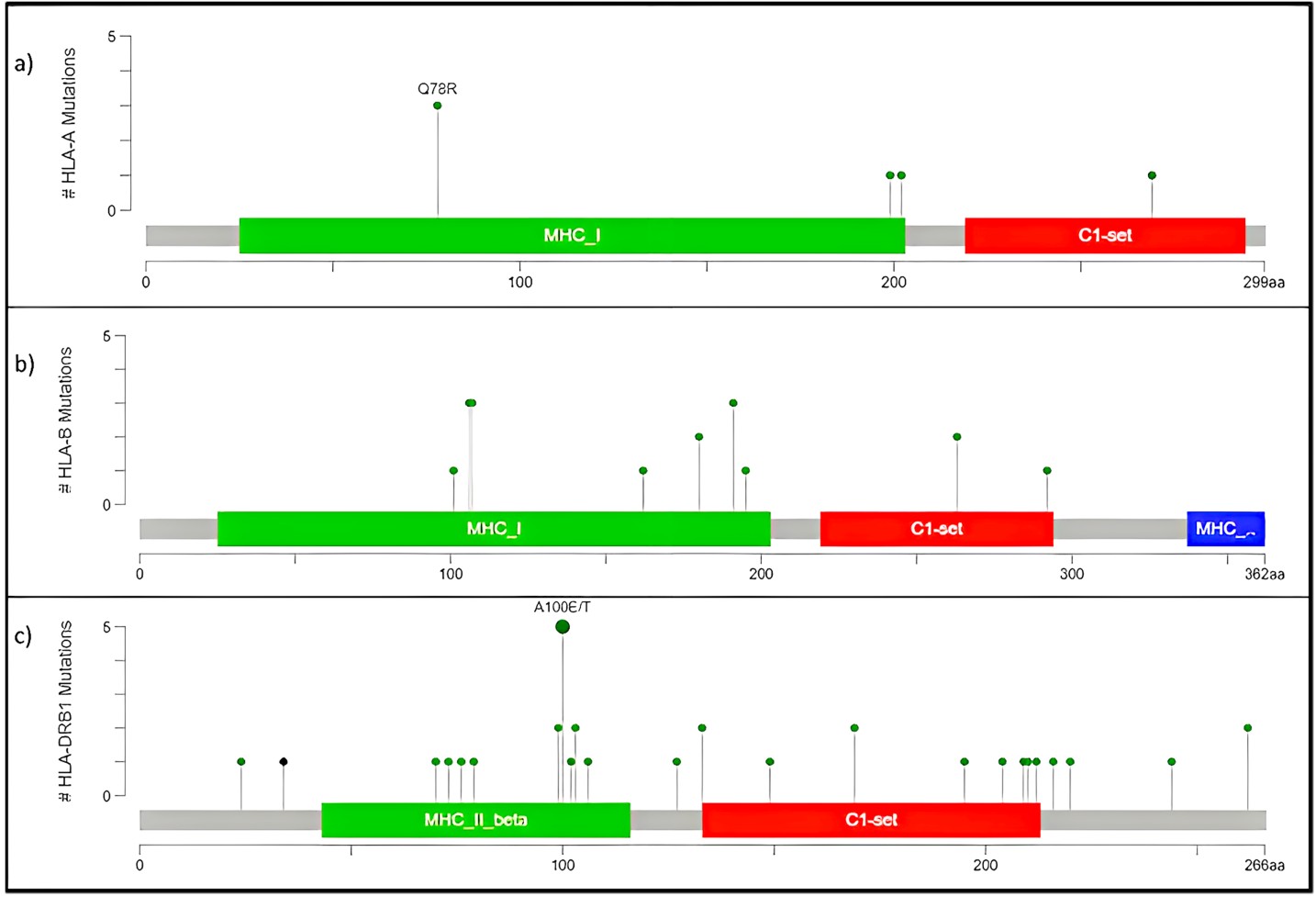

**Figure 6 Mutation map of missense variants (A) HLA-A, (B) HLA-B, (C) HLA-DRB1.** (A) HLA-A, has four mutations and three samples have a common point mutation Q78R, (B) HLA-B, (C) HLA-DRB1 has most no. of mutations, and single point mutation found in all affected samples A100E/T.                                               

(including breast cancer), identified by an algorithm described in *Chang et al. (2018)* and *Chang et al. (2016)* and is present in three out of four patient samples in this study (Fig. 3). The missense variant is mapped as shown in Fig. 7, causing a substitution of aspartic acid residue with alanine residue at the 74[th] position. SIFT (*StackPath, 2022*) and *PolyPhen-2 (2021)* predicted the functional impact of this variant to be deleterious. It is located in the Ankyrin repeat-containing domain (Prosite_profiles (*Hulo et al., 2006*)), a common protein-protein interaction motif.

### High-frequency somatic variant of ATF1

A nucleotide T deletion in the activating transcription factor 1 (*ATF1*) gene causes a splice region variant, which was observed in all samples reported by id rs4986837. The role of ATF1 in breast cancer has been identified as a tumor suppressor (*Huang et al., 2016*), including its activation by BRCA1, increasing transcription of MHC class genes and its
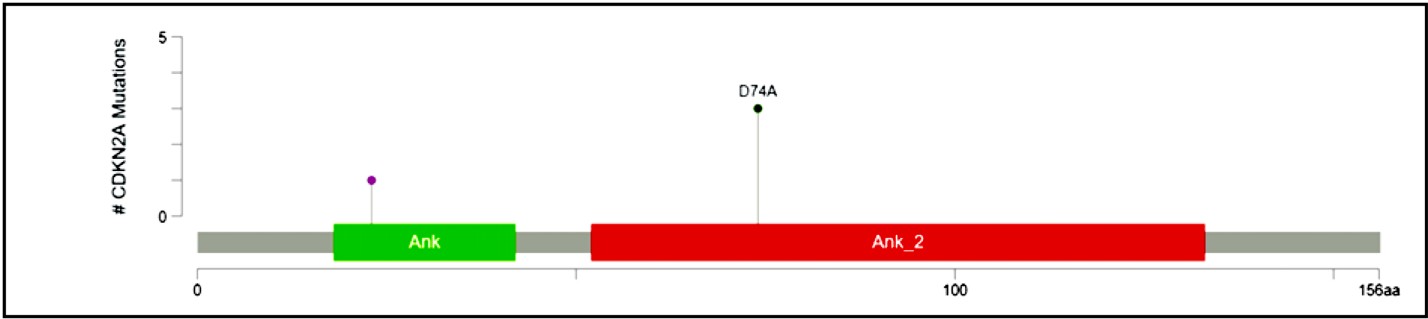

**Figure 7  Mutation map of CDKN2A.** A recurrent hotspot mutation rs200429615 resulting in a missense variant in the Ankyrin repeat-containing domain present in three samples.                                             

involvement in regulating the steroidal hormone synthesis (*Haakenson, Kester & Liu, 2012*). A splice region variant could be responsible for the loss of an exon or the inclusion of an intron resulting in an altered protein sequence. In this case, "GT", which is a splice donor site, is disrupted, suggesting the inclusion of intron further resulting in altered functions.

The observation that innate immune system and transcription factors were possible drivers of cancer in the patients analysed, we performed immune cell infiltration analysis of the tumors and matched normal tissue using CIBERSORTx (*Stanford University, 2023*).

## Tumor immune profiling of breast cancer subtypes

We generated gene expression profile all the samples using DESeq2 (*Love, Huber & Anders, 2014*; *Manjunath et al., 2022*) to analyse the differences in the immune cell fraction in individual samples and identify breast cancer subtype-specific patterns of immune infiltrating cells using CIBERSORTx (*Stanford University, 2023*). CIBERSORTx uses a gene expression deconvolution algorithm to estimate the relative proportions of 22 distinct functional subsets of immune cells. Among the ER+ve cancers, we observed that luminal A breast cancer showed the presence of CD8[+]T cells and NK cells activated, whereas luminal B showed negligible/ absence of CD8[+]T cells and high Tfh cells. Among the ER-ve group, Hmod showed a higher presence of B cells naïve, Tcells CD4[+]naïve, M2 macrophage and neutrophils and low Macrophage M1 and memory B cells. In TNBC, a higher fraction of T cells CD4[+]naïve, natural killer (NK) cells activated, T follicular helper (Tfh) cells, dendritic cells activated, and neutrophils compared to the matched normal was observed. A lower fraction of M1 macrophage was observed in TNBC (Fig. 8).

We observed the differential presence of myeloid and lymphoid cells in a subtype-specific manner. The missense mutations observed in the MHC-I and MHC-II in individuals were correlated to the presence of the immune cell types with focus on dendritic cells (DC) and Macrophages.

We further performed gene co-expression network analysis to correlate the BC subtypes with the immune microenvironment.

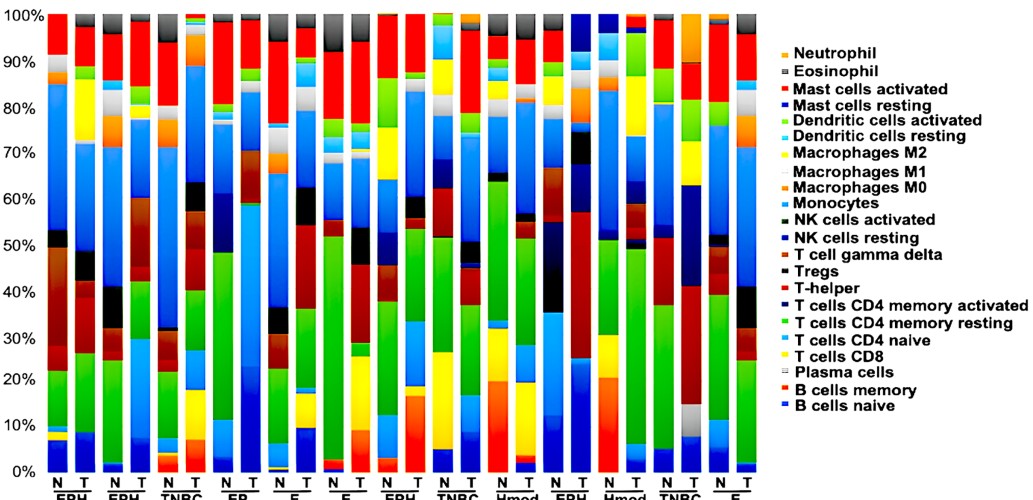

**Figure 8  Tumor immune profile of breast cancer subtypes using CIBERSORTx (*Stanford University, 2023*).** The plot shows distribution of immune cells across breast cancer subtypes in both tumor and normal samples. N, normal; T, tumor; ER, estrogen receptor; PR, progesterone receptor; HER2, human epidermal growth factor receptor2; EPH-ER, PR, HER2 positive; TNBC, triple negative breast cancer negative for ER, PR, HER2; EP- ER, PR positive; Hmod-HER2 positive; E-ER positive.

## Weighted gene co-expression network analysis (WGCNA) revealed subtype specific gene expression modules

We used the normalised count obtained from DESeq2 as input for WGCNA (*Love, Huber & Anders, 2014*; *Manjunath et al., 2022*). Figure 9 shows the modules which are significant and have high correlation with the BC subtype. In EPH, the MECyan module has the highest positive correlation of 0.66 and *p*-value < 0.01. Similarly, the MEgreen module is the most significant with 81% correlation in only estrogen positive BC. Among all the subtypes, the highest and most significant correlation was in the TNBC subtype (Table 2). We obtained the gene list from each of the modules and subjected it to STRING (*STRING, 2022*) pathway analysis. Table 2 shows the enrichment of immune-related pathways in all subtypes. In the EPH subtype, the pathway showed enrichment in the IL-25, IL22 and B cell markers CD19, BTK, (Fig. S2A and Table S1A) which are present at higher levels compared to the matched normal, and also correlates with the presence of higher B cells and Tfh cells. To find out if the genes in the pathway had mutations which can explain normal/abnormal functioning of the resident immune cells, we explored the exome of the EPH group. Interestingly, we found mutations in TLR10, BTK, CARD8 and C7, which are not connected but are mutated indicating both innate and adaptive immune genes are affected in EPH (Table S1A). In the Estrogen only positive, CD4+ve and Tfh cells were enriched with a pathway showing T-cell mediated immunity and the mutation in IL4R was observed (Fig. S2B and Table S1B). The number of genes mutated in the ER+ve tumors were the least. In contrast, ER-ve tumors, Hmod showed T gamma delta enriched and the pathway enriched was Type II immune response pathway (Fig. S2C), with mutations in

## Module−trait relationships

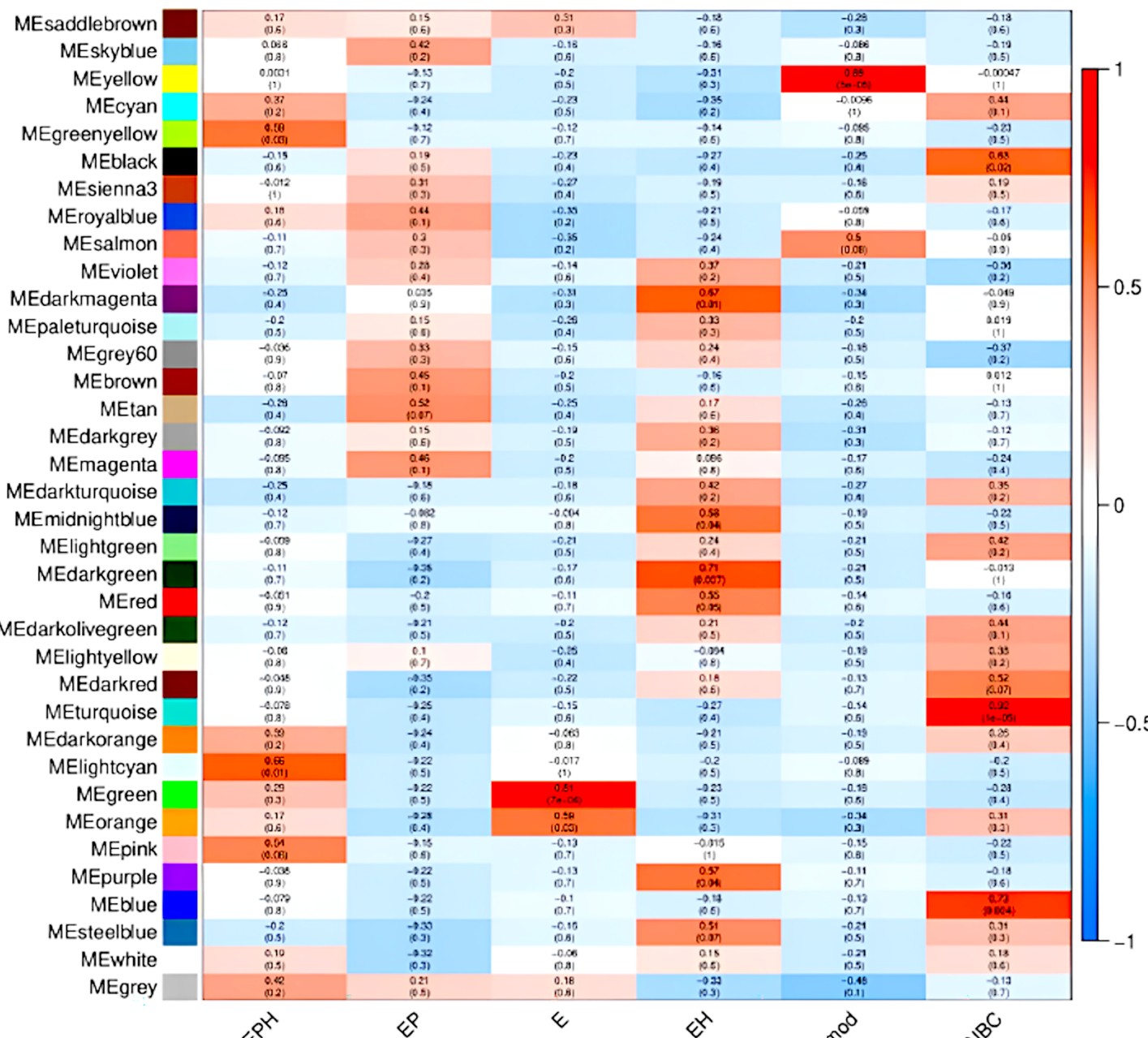

**Figure 9 Module trait relationships obtained by weighted gene co-expression network analysis.** Consensus network module correlated with hormonal status. correlation coefficient along with *p*-value in parentheses. Color coding is according to the correlation coefficient. ER, estrogen receptor; PR, progesterone receptor; HER2, human epidermal growth factor receptor2; EPH-ER, PR, HER2 positive; TNBC, triple-negative breast cancer negative for ER, PR, HER2; EP- ER, PR positive; Hmod-HER2 positive; E-ER positive.

CCL4, FCGR3B, GPX1 and IFNAR1 (Table S1C). TNBC was enriched in Tfh cells and neutrophils and the pathway was Interleukin and interferon pathway (Fig. S2D) with the largest number of mutations in the genes like CR1, C8A, NOS1, NOS2, ITGAM, IL5RA, ITGAM, NCAM1 and LTBP1 (Table S1D).

**Table 2 The pathways regulated in breast cancer subtypes as per the WGCNA module correlation.**

| BRCA type | $p$-value | Correlation value | Colour | Pathway |
|---|---|---|---|---|
| EPH | 0.01 | 0.66 | Light cyan | Immune system pathway and cytokine signalling |
| EP | – | – | – | – |
| E | 7e−04 | 0.81 | Green | T cell immunity |
| EH | 0.01 | 0.67 | Dark magenta | – |
| Hmod | 5e−05 | 0.89 | Yellow | Type II immunity |
| TNBC | 1e−05 | 0.92 | Turquoise | Interferon and interleukin pathway |

## Discussion

Breast cancer is known for its multifactorial and aggressive nature in advanced stages. Transcriptome study helps to understand this complexity by shedding light on regulatory networks involving many gene hubs and regulatory molecules. RNA-seq also enables isoform-level expression that might be involved in different pathways conducting different functions (*Wang et al., 2016*). Integrating exome-seq with RNA-seq helps in understanding somatic variation, meiotic recombination, cell-to-cell heterogeneity in gene expression and DNA-RNA regulation like abnormal splicing, which leads to aberrantly spliced transcripts (*Wang et al., 2018*; *Yamada et al., 2019*).

Cancer driver genes have somatic mutations that help tumorigenesis, whereas passenger genes carry neutral mutations that do not give cancer cells a selective advantage (*Dietlein et al., 2020*; *Martínez-Jiménez et al., 2020*). The driver mutations occur in functionally important genomic positions corresponding to amino acids critical for the protein function. They also happen in excess over the background mutability of the genome owing to positive selection in the tumor (*Dietlein et al., 2020*). Driver mutations in oncogenes lead to activating or new functions, whereas, in tumor suppressors, driver mutations are inactivating (*Agilent Technologies, 2021*; *Pon & Marra, 2015*). Powerful cancer genomics tools are thus required to identify these rare driver mutations among several passengers. DawnRank is one tool that can discover novel and personalised individual drivers based on the overall differential expression of its downstream gene in the molecular interaction network (*Hou & Ma, 2014*).

The driver genes with somatic variants in the study were *HLA-A, HLA-B, HLA-DBR1, C3, CDKN2A* and *ATF1*, which were related to two pathways viz innate immune system and cell cycle. HLA-A, HLA-B and HLA-DRB1 genes were upregulated in all patients, actually indicating a good prognosis since human leukocyte antigen (HLA) class I molecules play a vital role as antigen-presenting molecules for cytotoxic T lymphocytes (CTLs), and HLA class II molecules (HLA-DR and HLA-DQ) are essential for peptide presentation to T-helper lymphocytes (*da Silva et al., 2013*). Also, loss of HLA class I and class II molecules contributes to tumor aggressiveness, invasiveness, and metastatic potential due to escape from being killed by CTLs (*Kaneko et al., 2011*). However, though upregulated, integration of the exome study showed that all the HLA-related genes had missense mutations indicating that these TSGs might be driver genes with mutations

leading to inactivation. In addition, the mutations in HLA genes are located in the binding region, which might directly affect the innate or adaptive immune system.

C3 is an integral part of the complement system and was downregulated in all the patients in the study, indicating a good prognosis because the complement system is pathologically activated in the tumor microenvironment, which promotes tumorigenesis by regulating inflammation; stromal cell immunity; and the proliferation, epithelial-mesenchymal transition (EMT), migration and invasiveness of tumor cells (*Zhang et al., 2019*; *Niculescu et al., 1992*). The exome data revealed an intron variant in a sequence feature (a beta-strand) in C3, which might be involved in its regulation, further affecting the innate immune response.

ATF1, a reported tumor suppressor (*Huang et al., 2016*), carried a splice variant in all samples. It was downregulated in three and upregulated in one sample. But since it was mutated, its upregulation had no prognostic value. Similarly, CDKN2A is a cyclin dependent kinase inhibitor that suppresses cell proliferation (*Dębniak et al., 2005*). Though upregulated in one sample, the missense variant in *CDKN2A* was identified as a cancer mutation hotspot. This tells us that the integration of exome and transcriptome data is important to understand the exact impact of genes in cancer. This also suggests that the innate immune system and cell cycle might have a significant role to play in Indian breast cancer.

Cancer's well-known hallmarks are immune modulation, selective growth and proliferative advantage (*Fouad & Aanei, 2017*). Based on the pathway analysis, the innate immune system and cell cycle were the most altered considering the top 60 genes. The cellular network of innate immune system cells viz. granulocytes (neutrophils, eosinophils, and basophils), DCs, NK cells, myeloid-derived suppressor cells(MDSCs) and macrophages play an important role in antitumor immunity. The innate immune system recruits the adaptive immune system with the help of secreted cytokines to enhance antitumor immunity (*Gatti-Mays et al., 2019*).

The ER status guides the risk of relapse and chemotherapy response. In ER-ve tumors, the presence of CD+ve T cells and activated memory T-cells was associated with a reduced risk of relapse, and neoadjuvant chemotherapy response can be predicted based on the presence of T follicular helper cells (*Raza Ali et al., 2016*). In the Indian cohort of ER-ve samples, in TNBC, we observed high Tfh and high neutrophils. In Hmod, we observed the presence of gamma delta T($\gamma\delta$T) cells, indicating the difference in the immune profile of the TCGA cohort *vs* the Indian cohort. $\gamma\delta$T are unconventional T cells that migrate to peripheral tissues and function independently of major histocompatibility complex (MHC)-dependent antigen presentation (*Park & Lee, 2021*). $\gamma\delta$T exhibit strong cytotoxicity and hence are promising therapeutic targets for cancer immunotherapy (*Park & Lee, 2021*).

In ER +ve tumors, M0 macrophage in the TCGA cohort was associated with poor prognosis (*Raza Ali et al., 2016*); in the Indian cohort, ER +ve tumors showed the presence of T cell-mediated immunity, which coincides with the high CD4+ve naïve and memory cell and the presence of Tfh cells. In EPH tumors, B cells and Tfh cells were observed, indicating a possibility that patients might benefit from neoadjuvant chemotherapy.

The innate immune system has been studied in cancer, and recent research has even established the roles of cytokines in breast cancer progression and invasion (*Setrerrahmane & Xu, 2017*). Immunotherapies presently target only the adaptive response and not the innate response, which would be one reason for their effectiveness in only some patients. For this reason, it is essential to understand the effects caused by the innate Immune system and develop drugs targeting the same (*Sabry & Lowdell, 2020*). Additionally, innate immunotherapy is likely to improve the outcome because apart from being the first line of defence against cancer, and infection, 10% of the lymphocytes in the peripheral bloodstream are a part of innate immune response and 95% of those consist of natural killer cells (*Sabry & Lowdell, 2020*). Though nonspecific, the innate immune system can provide a rapid response in minutes or hours after aggression (*Sabry & Lowdell, 2020*; *Marshall et al., 2018*). Through this study, we could identify many mutations in breast cancer related to the Immune system, which would further help us realise the potential of innate response in Indian patients. All the stated mutations need validation, and their functional impact needs to be studied in many patient samples to better understand the mechanisms driving breast cancer in Indian patients.

## ACKNOWLEDGEMENTS

We thank Dr. Mallika Natraj, Dr. Rahul S Kanaka and the patients for providing the tumor samples. We are grateful to Professor Narayan Rao Yathindra for his constant guidance and support throughout the study. We acknowledge the IBAB sequencing facility, Dr. Hosahalli Subramanya for the support and Meghana Manjunath for procuring samples.

### Funding

This work was supported by grants from the Department of Science and Technology (SR/FST/LSI-536/2012) and the Department of Biotechnology (BT/PR13458/COE/34/33/2015). Snehal Nirgude was supported by DST-INSPIRE (Ref. no. IF140949/2015, Innovation in Science Pursuit for Inspired Research, Dept. of Science and Technology, Govt. of India). The funders had no role in study design, data collection and analysis, decision to publish, or preparation of the manuscript.

### Grant Disclosures

The following grant information was disclosed by the authors:
Department of Science and Technology: SR/FST/LSI-536/2012.
Department of Biotechnology: BT/PR13458/COE/34/33/2015.
DST-INSPIRE: IF140949/2015.
Innovation in Science Pursuit for Inspired Research.
Dept. of Science and Technology, Govt. of India.

### Competing Interests

The authors declare that they have no competing interests.

## Author Contributions

- Snehal Nirgude conceived and designed the experiments, performed the experiments, analyzed the data, prepared figures and/or tables, authored or reviewed drafts of the article, and approved the final draft.
- Sagar Desai performed the experiments, analyzed the data, prepared figures and/or tables, authored or reviewed drafts of the article, and approved the final draft.
- Vartika Khanchandani conceived and designed the experiments, performed the experiments, analyzed the data, prepared figures and/or tables, and approved the final draft.
- Vidhyavathy Nagarajan conceived and designed the experiments, performed the experiments, analyzed the data, prepared figures and/or tables, and approved the final draft.
- Jayanti Thumsi analyzed the data, authored or reviewed drafts of the article, provided samples and clinical details, and approved the final draft.
- Bibha Choudhary conceived and designed the experiments, analyzed the data, prepared figures and/or tables, authored or reviewed drafts of the article, and approved the final draft.

## Human Ethics

The following information was supplied relating to ethical approvals (*i.e.*, approving body and any reference numbers):

The BGS Global Hospitals and IBAB approved this study (IEC/Approval/2018-05/06/01A).

## Data Availability

The sequence reads are available at GenBank: PRJNA835602.

## Supplemental Information

Supplemental information for this article can be found online at http://dx.doi.org/10.7717/peerj.16033#supplemental-information.

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
