# Peer review of "Integration of exome-seq and mRNA-seq using DawnRank, identified genes involved in innate immunity as drivers of breast cancer in the Indian cohort"

_PeerJ, doi:10.7717/peerj.16033_

## Round 0.1 · original submission · Minor Revisions

We have received two reviewing comments suggesting minor revision. There are numerous remarks. Please carefully check the Figures quality.

Reviewer 1 ·

Basic reporting

Minor comment: typo in line 144 beginning in "Here we analysed the data for samples that had both exome-seq and mRNA-seq data.ore than.."
Figure legends must include more information. Such as: PCA plot for Normal and tumor samples based on RNA-seq etc.
Figure 4 missing labels for a and b. Also, Figure 4 is mentioned as waterfall plot, which is not correct
Figure 8 legend is not described sufficiently

Experimental design

No comment

Validity of the findings

No comment

Additional comments

It would be helpful to show heatmap of differentially expressed genes from RNA-seq data

Reviewer 2 ·

Basic reporting

The authors of the manuscript titled "Integration of exome-seq and mRNA-seq using DawnRank, identified genes involved in innate immunity as drivers of breast cancer in the Indian cohort" reports the involvement of the innate immune system in breast cancer specifically among Indian patients.
Although the objective of the manuscript is very much of importance and significance and authors seem to have a good strategy to carry out the computational framework, the reporting of the work needs a bit more improvement. I will go through them point wise

1. The sentences are confusing in a couple of places for example- line 19-20
"We identified CD8+ve T cells, M2 macrophages, and neutrophils to be enriched in luminal A and triple-negative breast cancer(TNBC) subtypes, respectively"
Here authors mentioned three immune cell types -CD8+ve T cells, M2 macrophages, and neutrophils to be enriched in two sample subtypes "respectively".

2. Authors have used abbreviated form of some keywords without mentioning full form on first appearance.

3. I would highly suggest the authors to avoid the overstatement in the manuscript. For example- line 331- "Also the innate immune system and cell cycle have a significant role to play in Indian breast cancer"

4. Some statements require more citations for example line 42-
"One of the hallmarks of cancer progression is the evasion of the immune system".
It has only one citation.

5. Figure quality is extremely poor. Some plots have cropped labels. All the plots need to be better quality wise so that it is visible clear , crisp and legible.

Experimental design

The authors have done a good job in framing the analysis but I have some concerns pointed below-

1. The authors did not mention the assembly of human genome reference in line 98. (for example if it was - GRCh38.p13)

2. It is not clear from line 98-100 -"The adapter trimming was done, followed by alignment and coverageBed from BEDTools was used to extract the count per transcript per sample using the annotation files" , what authors want to convey. The sentence formation is flawed and lost in translation. Should be clear.

3. The authors have used DESeq R package for differential expression analysis of normal and tumor patients. I wonder why not DESeq2 ?

4. The authors did not mention the parameters used in preprocessing of the data which should be mentioned in the manuscript. For example the authors have mentioned FastQC tool but gave no information or graphical representation of quality check at all. There is no information provided for basic statistics or phred score used or other parameters.

5. The authors simply mentioned "The reads were further processed and aligned, and variants were called as detailed in the study mentioned" and cited the study. I believe the authors should feed the details about the program used for the variant calling with proper information of parameters used.

6.Point 5 also goes the same for section 2.7 of the manuscript. There is absolutely no informative details of the integration and its parameters except for saying that DawnRank was used. It would be very helpful for the readers if the authors present a graph/plot or more detail in methods section about DawnRank.

Validity of the findings

The output of the study enlightens the breast cancer research but I would suggest the authors to use words like "seem to impact" or "potential drivers" when they talk about the genes. Since it is a pure computational study , I would not recommend the authors to sound overstated.

The results and discussion parts of the manuscript are interesting and well stated except for some major pitfalls at a couple of places.

1. Result section 3.1 states- "Transcriptome Analysis led to the identification of the altered pathways in Breast Cancer subtypes". The details under this section does not really talk about altered pathways at all.

2. The same section 3.1 also lacks any volcano or MA plots to show the differential expression of the found genes. Since DESeq was used , I fail to understand why there is no graphical information of the analysis when it is very straightforward from this R package.

3. The same section 3.1 states at line 152 "No common gene was obtained among the four samples" which by the venn diagram as well as by the very next lines of the authors seem to be contradictory.

4. Fig 1b- There are two EPH labels (EPH1 and EPH2). It would be nice to relabel them with some kind of subtypes identifier for both of these EPH samples, instead of just numbering them.

5. Line 153- I would suggest to also say you are talking about samples P1 and P2 .

6. Please put reference in line 215-217.

Additional comments

I overall am in the favor of the manuscript except for the major points I made. Discussion part is well written and complies the information well. I would still recommend the authors to go over the text again to fix the overstatements and sentence errors. The figure quality really needs to be improved.

---

## Round 0.2 · Minor Revisions

The reviewers have no more critical remarks. But the manuscript has some technical issues to be fixed.

I believe it needs minor revision without an additional reviewing round.

Please see the reviewer’s comment about Figures 3, 4 and 7.
The figure quality is not enough. Please rearrange the figure panel, put the legend below. May fit the figure width to the page width, then the legend and designation on the histograms will be readable.
It is small and distorted font size now.

As the Academic editor I have some other comments to be fixed in the revised version.
There are duplicated (possibly due to citations or web-links)
Line -
176 tool CIBERSORTx(ìCIBERSORTxî). – it i

Please remove color marks -
Line 192
Here…

See duplicated reference for Seseq -
Line 197
tumour samples. Using DEseq(Anders & Huber, 2010)(ìDESeqî)…

Line 423
In Hmod, we observed gamma delta T[ ¬F ] cells –0 Greek letters lost. Please check font

Line 444
…and 95% of those consist of natural killer cells(Lowdell). – the reference ‘Lowdel’ is not in place, or not complete
See line 446 – not complete or extra reference
aggression(Lowdell)(Marshall et al., 2018).

In Acknowledgments section (after line 457)
“We also..” – duplicate wording

Reference in line 476 Begum SA… - remove all uppercase capital letters, make standard title for this reference…

Line 519-520: DESeq. Available at http://bioconductor.org/packages/DESeq/(accessed January 9,
2021). – please update the access date by 2023

Same for the reference in line 527 Extraction of DNA from TRIZOL preparations. (fic more recent access date)
Same for line 571 Lowdell M…. --- check the access date
Reference in
542 Hou JP. 2016. – not enough reference data (age numbers, URL?)
* * *
The remarks above rather technical. More important to provide publication quality figures, not blurred.
Please check again the formatting (add space before parentheses through the text)
Waiting for resubmission of the revised manuscript.

Reviewer 1 ·

Basic reporting

Authors have addressed all comments

Experimental design

No comments

Validity of the findings

No Comments

Reviewer 2 ·

Basic reporting

Imprved

Experimental design

Improved

Validity of the findings

Figure 3, 4 and 7 are either blurred or too stretched and quality of all figures still need much improvement. I wonder if they are exported in high quality after they are produced. Kindly look into it.


Rest, manuscript has been improved substantially and acceptable now.

Additional comments

None

---

## Round 0.3 · accepted · Accept

Thanks for the updates. No more critical remarks. Accept manuscript for publication.

The Section Editor noted that some minor editing is required. For instance, spaces are needed to separate parentheses from the previous text. Lines 358-360, edit refer.

Reviewer 2 ·

Basic reporting

i thank the authors for incorporating all the changes regarding figure quality and overall manuscript is much improved now and suitable for publiication.

Experimental design

Good

Validity of the findings

Satisfactory